# Application Study on Double-Constrained Change Detection for Land Use/Land Cover Based on GF-6 WFV Imageries

**Jingxian Yu [1,2], Yalan Liu [1,*], Yuhuan Ren [1], Haojie Ma [1,2], Dacheng Wang [1], Yafei Jing [1,2] and Linjun Yu [1]**

[1]   Aerospace Information Research Institute, Chinese Academy of Sciences, Beijing 100101, China; yujx@radi.ac.cn (J.Y.); renyh@radi.ac.cn (Y.R.); mahj@radi.ac.cn (H.M.); wangdc@aircas.ac.cn (D.W.); jingyafei19@mails.ucas.edu.cn (Y.J.); yulj201831@aircas.ac.cn (L.Y.)

[2]   University of Chinese Academy of Sciences, Beijing 100049, China

*   Correspondence: liuyl@aircas.ac.cn; Tel.: +86-139-1103-2598

**Abstract:** As a new satellite sensor of the GaoFen (GF) series, GF-6 Wide Field of View (WFV) with the resolution of 16 m has the characteristics of wide coverage, high-frequency imaging and has four new bands of two red-edge, yellow, and purple compared with GF-1 WFV. In order to test the validity of the supplementary bands of GF-6WFV data for change detection of land use/land cover (LULC), this study applied the Double-constrained Change Detection Method (DCDM) that uses the double constraints (change vector intensity and correlation coefficient) for change detection on object-level. According to two GF-6WFV imageries acquired in the Xiong'an New Area in June of 2018 and 2019, feature analysis was performed to determine whether the new bands are helpful to detect the change of LULC first. Then, by coupling these selected features, the intensity of change vector and correlation coefficient were used as the double constraints to perform the change detection. The study demonstrates that the relevant features of the two red-edge bands can achieve the overall accuracy of 89% for change detection of LULC and improved by 2% comparing with using the corresponding temporal GF-1WFV data, while the purple and yellow bands cannot provide enough effective information for this detection. This study can provide theoretical support for the in-depth applications of GF-6 WFV data products in the change detection fields and has explored its applicability and potential in resource and environment monitoring, it is helpful to the further applications.

**Keywords:** land use/land cover (LULC); GF-6 WFV; object-oriented; change detection; double constraints

## 1. Introduction

Land use/land cover (LULC) is a combination of surface elements covered by natural land and artificial construction structures. LULC change is a widely concerned issue in global environmental change and sustainable development presently. The urbanization in China has promoted the increase in land use/cover change during recent 10 years due to the national economic and social development, while the growing needs of implementation of land planning, comprehensive land management, cadastral management, farmland protection, law enforcement, and supervision for the LULC updating information is more and more urgent. Due to the short revisiting period, macroscopic and wide acquisition coverage, remote sensing technology is widely applied to the field of environmental monitoring by the related departments of China [1,2].

Change detection is the approach to detect the surface changes over time by remote sensing images acquired at different times for the same interesting area. The premise is that these changes will lead to Digital Number (DN) changes in the corresponding area between different images, and the

DN changes caused by real changes in ground truth is far greater than those caused by other factors, such as spectral changes caused by the observation angle, solar altitude angle, and atmospheric conditions at different times [3]. The basic approaches for change detection are direct comparison and post-classification comparison [4]. The former is to process the preprocessed original image through image enhancement, image transformation, or to use specific operators to extract image features, and compare the features of each image at the corresponding position to obtain the result of change detection [5]. The latter is based on the classifications for different imageries. It excessively depends on the accuracy of classification, therefore, its results for change detection are often unstable. In contrast, the results of the direct comparison method are more stable [6]. Bruzzone and Prieto first proposed an unsupervised change detection method based on the difference graph in 2002 [7]. Many subsequent researches on change detection, such as Change Vector Analysis (CVA) [8–12], have been based on their studies. As an extension of the difference method, CVA combines the difference value of bands between images into a change vector and uses the change vector to measure the change between bi-temporal images. The change intensity is obtained by determining the Euclidean distance between two data points in the N-dimensional space. Yang et al. conducted a comparative study on CVA and other commonly used methods; their results show that CVA has the lowest probability of omission [13]. Actually, CVA has been widely used in change detection of LULC, the main reason is that it is more suitable for change detection based on remote sensing images in terms of that it can use more or even all the band information to detect changing pixels [14]. However, this method also has shortcomings, that is, it is difficult to determine the change threshold.

According to this, Wang et al. of our team proposed the object-oriented Double-Constrained Change Detection (DCDM) model that combines CVA with correlation analysis between different objects to conduct change detection in 2018 [15]. Correlation analysis studies the degree of correlation between objects in multi-temporal imageries by calculating their correlation coefficient. Obviously, the greater the degree of correlation between an object in images of different time phases, the less likely it is to change. Therefore, using the dual constraints of the correlation coefficient and CVA can better determine the change threshold and reduce more errors and omissions.

At the same time, based on an object-level, DCDM can reduce the "salt and pepper" phenomenon compared to the pixel-level method and improves the detection accuracy [16–20]. However, these researches only based on the images of GaoFen (GF)-1 Wide Field of View (WFV) [21,22].

The GF-6 satellite, also known as the "High-resolution Land Emergency Monitoring Satellite", was launched on 2 June 2018. It is equipped with two cameras, one is the panchromatic/multi-spectral camera with high-resolutions of 2 m and 8 m, and the other is the WFV camera with 16 m multi-spectral medium-resolution and 800 km wide-coverage of observation. The WFV can provide great support for the investigations of agriculture, forestry, land and resources, disaster emergency, and other industries in China [23]. The parameters of its sensors include 4 more bands of purple, yellow, and two red-edge bands than those of GF-1WFV.

The above researches show that use object-level CVA and correlation analysis as double constraints to conduct change detection can obtain higher detection accuracy, so in order to study whether the 4 new added bands of GF-6WFV data are suitable for change detection, we used the object-oriented DCDM to do research on GF-6WFV data. The aim of the study is to provide a theoretical basis for the in-depth application of GF-6 WFV data products in change detection fields and explore the applicability and the potential of GF-6 in resource and environment monitoring [24]. We used the significance analysis of spectral, texture, and shape features for each band of GF-6WFV. Based on optimal features selection, the change vector intensity of objects after image segmentation for bi-temporal imageries and the correlation coefficient between the objects were used for coupling features to realize the automatic extraction of the change information. Finally, the comparison test was conducted with a pair of same time-phrase of GF-1WFV data to evaluate the result of GF-6 data.

## 2. Materials and Methods

### 2.1. Analysis of Band Spectral Characteristics

When detect changes for ground features from satellite remote sensing data, the values of features and the spectral characteristic curves are based on the multi-spectral information. Consequently, to make full use of this information for subsequent analysis, one needs to perform a holistic statistical analysis of the information to derive the spectral statistical characteristic values of each band, such as the maximum, mean, standard deviation, entropy, etc., for providing the basis for the subsequent feature selection and various types of analysis [25].

The remote sensing images can be expressed in three dimensions of $x$, $y$, and $z$, where $x$ and $y$ are used to represent plane space, and $z$ is band information, as shown in Expression (1).

$$I = f\,(x, y, z) \tag{1}$$

According to the needs of different analyses, it can be expressed by following three means:

(1) The image space $I = f\,(x, y, 0)$, simply and directly describes the spatial distribution of ground objects in the image and the relationship between spectral response and spatial position, but it is not sensitive to the relationship between bands.

(2) The spectral space $I = f\,(x, 0, z)$, it mainly reflects a spectral curve corresponding to the average radiation value of each pixel and depicts the characteristics of the electromagnetic wave energy with the wavelength change, that is, the spectral characteristic. The spectral response is the function of wavelength. Because of the difference in brightness between different objects as well as the same objects in different bands, these constitute the spectral feature information of the ground objects and make it possible to use the spectral space for image analysis and interpretation.

(3) The feature space $I = f\,(0, y, z)$, is a two-dimensional feature space in which the radiation intensity values of different features in two bands are plotted on a two-dimensional plane.

Statistical analysis was conducted on the 8 bands of GF-6WFV images to derive the related statistical characteristic values such as mean, maximum, and standard deviation.

### 2.2. Feature Analysis and Optimization of LULC

The process of the change detection for multi-temporal remote sensing images is to extract change information for the ground objects from the region of interests by the change detection algorithm and to obtain the changes through analysis and description. The core of change detection is to determine whether the features in the smallest detection unit have changed by the quantification of the difference between the features of this unit in different temporal images. Therefore, the selection of features is crucial [26]. Feature analysis was used to verify the influence of the new additional bands of GF-6WFV on the accuracy of change detection of LULC.

The surface features of remote sensing imagery can be described by spectral features, texture features, and spatial features. In this paper, these three types of features are selected for the quantitative analysis of objects in different temporal remote sensing imageries. The spectral features, often refer to the reflectance spectral characteristics, can be expressed by single-band grayscale mean, standard deviation, and band-specific feature index, such as Normalized Difference Vegetation Index (NDVI) and Normalized Difference Water Index (NDWI). The textual features describe the repeated local patterns in the image and their arrangement and include Angular Second Moment (ASM), dissimilarity, and correlation [27]. The shape features describe the contour features of objects in the image, which nclude aspect ratio, shape index, and area index. Its feature analysis can be carried out according to the classes of construction land, vegetation, bare land, and water body.

$$NDVI = (NIR - R)/(NIR + R) \tag{2}$$

$$NDWI = (Green - NIR)/(Green + NIR) \tag{3}$$

Green, R, NIR are B2, B3, and B4 bands in GF-6WFV data, respectively (See the next section for details).

The spectral feature curves are used to represent the different features of objects in spectral space. Therefore, it is necessary to select the valid features and remove the redundant and invalid features to improve the efficiency for change detection.

Feature selection is to select N (N ≤ M) features from the given M features to optimize the features, which is the key step for the pre-processing of change detection [25,26]. The purpose of this study is to select the features that are sensitive to LULC changes from the spectral and textural features of the images and to apply the selected features to the change detection model for reducing the dimension of data and to keep the original useful information to effectively describe the characteristic of changes for all the classes to the maximum extent.

In order to test the significance of the difference between the features for selected categories in each band, a test of significance approach was used to realize the optimization of features. The test of significance approach is used to test whether there is a difference between the effects of the experimental group and the control group or two different treatments and whether this difference is significant. The more significant the difference between classes in a certain feature, the more effective the feature can distinguish these classes, then it can be as the optimal feature. In this paper, Variance Analysis is adopted for this significance test. Its formula is as follows:

$$F = MS_b/MS_w \tag{4}$$

where $F$ is the statistic calculated by Analysis of Variance (ANOVA), $MS_b$ is the mean square between groups, and $MS_w$ is the mean square within a group [28].

When performing ANOVA on a feature, each $F$ value has its corresponding $p$-value. The larger the $F$ value and the smaller the $p$-value, the smaller the possibility of accepting the null hypothesis, that is, the difference in feature between groups is large. However, when the result of ANOVA is significant, the feature has a significant difference between the classes, but it does not indicate whether there are significant differences among all the classes under this feature or only among some classes.

In this case, multiple comparisons are required, that is, pairwise comparisons to determine the significance of each feature level. In multiple comparisons, if the calculated $p$-value is less than 0.05 at a given confidence level, the null hypothesis is rejected, and the two feature types are significantly different under this feature. Otherwise, the difference is not significant.

The confidence level does not indicate the size of the difference between groups, but it can show whether the difference between the two test groups is statistically significant. According to the needs of feature selection, one-way ANOVA with single-factor and multiple comparisons are taken to analyze the significant differences of various features between different classes, and the optimal features are selected after comprehensive analysis. In this paper, we used a 95% confidence interval (CI), that is, the $p$-value is equal to 0.05.

According to the above principles, the most suitable features for change detection in GF-6WFV data were selected.

### 2.3. Double-Constrained Change Detection

The "double-constrained" here refers to obtaining the change information by two thresholds through the change vector intensity and the correlation coefficient of objects in two-phase images. The double-constrained change detection is performed after image segmentation and feature optimization. The process is shown in Figure 1, and the specific methods are described as follows.

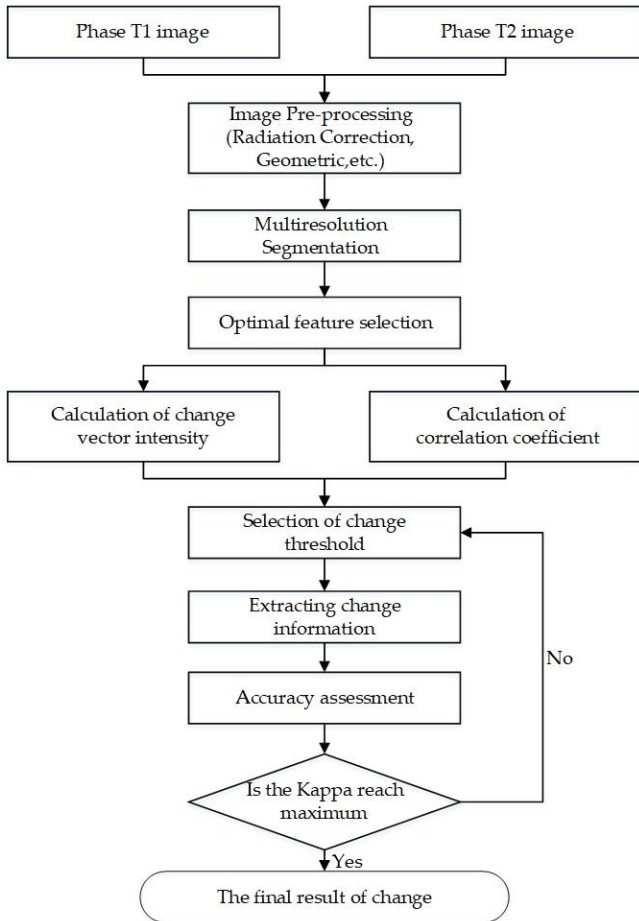

**Figure 1.** Flow chart of object-level double-constrained change detection (DCDM).

CVA is based on the radiation changes between images of different time phases, focusing on the analysis of the differences in each band to determine the intensity and direction characteristics of the change. Each pixel can generate a change vector with two characteristics of change direction and change intensity. The intensity of change is the Euclidean distance between two data points in n-dimensional space [5,7,10], its calculation is shown in Formula (5).

$$\|\Delta G\| = \sqrt{\sum_{k=1}^{n} (g_k - h_k)^2} \tag{5}$$

$\|\Delta G\|$ is the intensity of the change vector. The larger the value is, the greater the possibility of the change is. Conversely, the less the probability of the change is. Under normal circumstances, the initial threshold of the intensity of the change vector is set to 0.8, which needs to be adjusted appropriately according to different regions. $g_k$ and $h_k$ represent the features in the images of T1 and T2 phases, respectively, and $n$ is the number of selected features.

Generally, it is not accurate enough to use CVA alone to express the features of changes, so the correlation coefficient was used to further measure the possibility of change together with CVA in this study. The correlation coefficient is the result of correlation analysis in change detection and can be used to evaluate the correlation between the same object in different time phases images. The larger the correlation coefficient, the greater the correlation of the object between two images and the less likely for them to change. The addition of the correlation coefficient can better determine the change



threshold and reduce more errors and omissions. The correlation coefficient can be calculated by Formula (6):

$$R = \frac{\sum_{k=1}^{n}\left\{\left[x_{i1}^k - \overline{x}_{i1}\right]\cdot\left[x_{i2}^k - \overline{x}_{i2}\right]\right\}}{\sqrt{\sum_{k=1}^{n}\left[x_{i1}^k - \overline{x}_{i1}\right]^2 \times \sum_{k=1}^{n}\left[x_{i2}^k - \overline{x}_{i2}\right]^2}} \tag{6}$$

where $R$ is the correlation coefficient, the objects with small correlation coefficients are less likely to change. $n$ is the number of selected features, which $x_i^k$ represents the average DN of the $i$-th object in the k-band in a certain time-phase image and $\overline{x}_i$ represents the average DN of the n-band of the $i$-th object in a certain-phase image.

Through the above data analysis, the results are applied to the Double-constrained Change Detection Model. The first process is the multi-scale segmentation for the input images, the second is the optimal features selection for each object, the third is the calculation of the correlation coefficient and the intensity of change vector, and the last is to extract the change areas in the image using the double constraints.

### 2.4. Accuracy Assessment

A confusion matrix was used to evaluate the detection accuracy of the above change detection method. The results are as shown in Table 1. Each column of the confusion matrix represents the predicted categories, and the total number of each column represents the number of samples predicted to be in the appointed category. Each row represents the actual category, and the total number in each row represents the number of instances in a certain category.

**Table 1.** Confusion matrix of change detection for land use/land cover (LULC).

| Evaluation Data \\ Detection Result | Unchanged | Changed | Total |
|---|---|---|---|
| Unchanged | $N_{nn}$ | $N_{cn}$ | $N_{tn}$ |
| Changed | $N_{nc}$ | $N_{cc}$ | $N_{tc}$ |
| Total | $N_{nt}$ | $N_{ct}$ | $N$ |

Some parameters, such as commission errors, omission errors, precision, and overall accuracy, are usually applied to assess the detection accuracy.

Commission Errors = $N_{cn}/N_{tn}$ represents the proportion of samples in which unchanged categories are detected as changed categories;

Omission Errors = $N_{nc}/N_{tc}$ represents the proportion of the undetected samples that are the actual changes;

Precision = $N_{cn}/N_{ct}$ represents the proportion of the real change samples in all detected changes;

Overall Accuracy = $(N_{nn} + N_{cc})/N$ represents the proportion of samples correctly detected;

Kappa is used to measure classification accuracy; the result of performing a KAPPA analysis is a KHAT statistic (an estimate of KAPPA), its formula is as follows:

$$k_{hat} = \frac{N\cdot(N_{nn} + N_{cc}) - (N_{tn}\cdot N_{nt} + N_{tc}\cdot N_{ct})}{N^2 - (N_{tn}\cdot N_{nt} + N_{tc}\cdot N_{ct})} \tag{7}$$

Among them, $N_{nn}$ represents the number of samples that have not changed both in the test results and the reality, $N_{cn}$ represents the number of samples that have been erroneously detected as unchanged, $N_{tn}$ represents the total number of samples that have not changed in the test results, and $N_{nc}$ represents the number of unchanged samples that have been erroneously detected as changed, $N_{cc}$ represents the number of samples whose detection result is changed and is in line with the actual situation, $N_{tc}$ represents the total number of changed samples of the detection result, $N_{nt}$ represents

the total number of actually unchanged samples, $N_{ct}$ represents the total number of actually changed samples, and $N$ is the total number of samples.

The specific change detection process is shown in Figure 1.

## 3. Study Area and Data

The Xiong'an New Area is a state-level new area established in Hebei, China on 1 April 2017, and covers Xiongxian, Rongcheng, and Anxin counties. It serves to further promote the coordinated development of the Beijing-Tianjin-Hebei region and relief the nonessential functions of Beijing in an orderly fashion [29]. After the Xiong'an New Area was started following the Shenzhen Special Economic Zone and Shanghai Pudong New Area, the urban-rural, industrial and mining, and residential land in this new area has increased substantially and has derived its land use and cover being undergone obvious changes. Therefore, this is very suitable as a study area for change detection of LULC. The studied area for this study is located between 38°54′22′′–39°9′11′′ N and 115°42′50′′–116°13′26′′ E and occupies a land area of 1183.6 square kilometers, shown in Figure 2. In the selected area, there are mainly four types of ground objects: vegetation, construction-land, bare land, and water bodies. Among them, vegetation covers the largest proportion, followed by water bodies, built-up areas, and bare land.

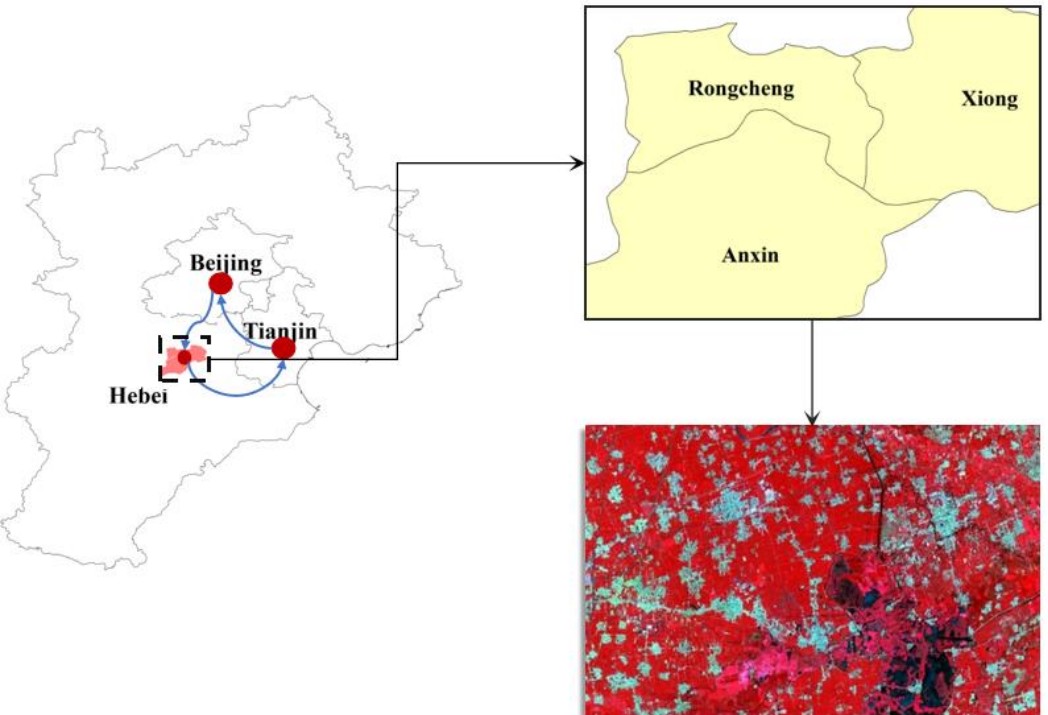

**Figure 2.** The study area.

The study is based on bi-temporal data obtained by GF-6WFV on 6 September 2018 and 6 September 2019 shown in Figure 3, and data obtained by GF-1WFV on 24 September 2018 and 24 September 2019, and their spatial resolution is 16 m shown in Figure 4. A comparative test was conducted on the two sets of data to study the usefulness of the newly launched GF-6WFVdata in the field of change detection.

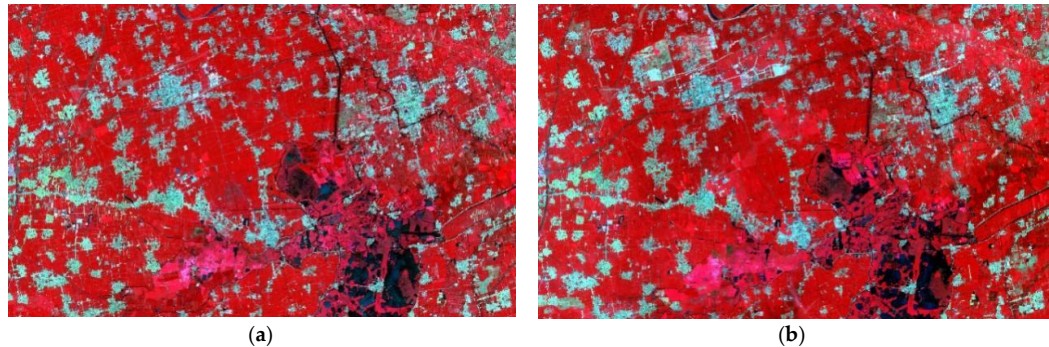

**Figure 3.** Two GF-6 Wide Field of View (WFV) images for the study area ((**a**) is for 6 September 2018, (**b**) is for 6 September 2019).

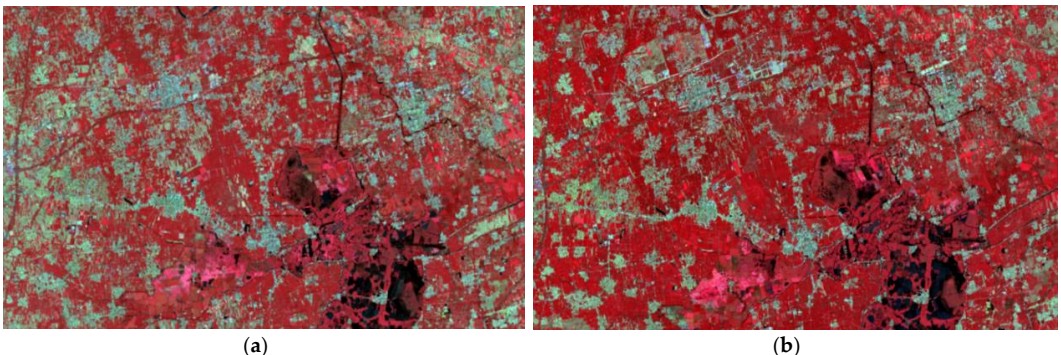

**Figure 4.** Two GF-1WFV images for study area ((**a**) is for 24 September 2018, (**b**) is for 24 September 2019).

GF-6 has the characteristics of wide coverage, high-quality imaging, high localization rate, etc., and has a breakthrough in the super-large field-of-view imaging technology in a single projection center, which improves the accuracy of wide-field camera images [30]. Its specific parameters of sensors are shown in Table 2. The additional red edge bands of GF-6 WFV comparing with GF1 WFV can effectively reflect the unique spectral characteristics of crops [23]. In theory, it can provide a greater amount of information for monitoring crop growth, and it has good application potential for the detection of changes in vegetation-covered areas.

**Table 2.** Technical indicators of sensors payload for GaoFen (GF)-6.

| Parameters | | Cameras | | | |
|---|---|---|---|---|---|
| **Spectrum (µm)** | panchromatic | 0.45–0.90 | | | |
| | multi-spectral | B1 (blue) | 0.45−0.52 | B7 (purple) | 0.40–0.45 |
| | | | | B1 (blue) | 0.45–0.52 |
| | | B2 (green) | 0.52–0.59 | B2 (green) | 0.52–0.59 |
| | | | | B8 (yellow) | 0.59–0.63 |
| | | B3 (red) | 0.63–0.69 | B3 (red) | 0.63–0.69 |
| | | | | B5 (red-edge1) | 0.69–0.73 |
| | | B4 (near-infrared) | 0.77–0.89 | B6 (red-edge2) | 0.73–0.77 |
| | | | | B4 (near-infrared) | 0.77–0.79 |
| **Spatial resolution (m)** | panchromatic | 2 | | 16 | |
| | multi-spectral | 8 | | | |
| **Width (km)** | | 90 | | 800 | |

## 4. Results

### 4.1. Feature Analysis and Optimization

Before change detection, the GF-6WFV images in two different years were preprocessed by radiation correction and geometric correction, and then the images were segmented by object-oriented multi-scale [31–33]. After multiple experimental comparisons, a set of optimization segmentation parameters was selected, including segmentation scale of 40, shape factor of 0.1, and compactness of 0.9. Through these processes, a more homogeneous patch can be obtained, that is, the segmentation result is not too broken and reflects the difference information between the patches, making the interior of the object more homogenous and is consistent with the actual boundaries of the features. Figure 5 shows two sets of tiles taken from the study area after multi-scale segmentation.

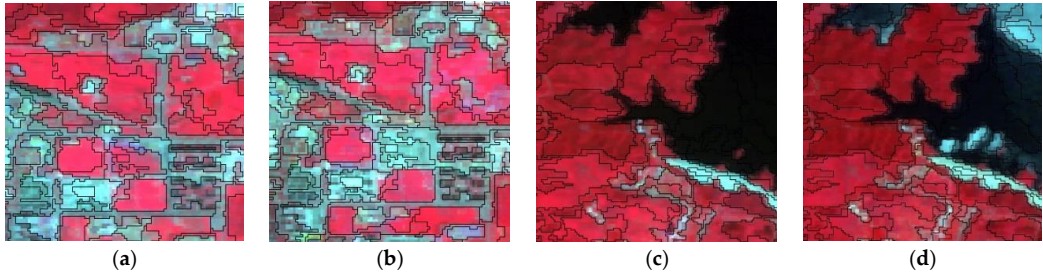

| (a) | (b) | (c) | (d) |

**Figure 5.** Segmentation tile diagram for GF-6WFV (The segmentation scale is 40, and the shape factor and compactness are 0.1 and 0.9, respectively). (**a**,**b**) represent tile in the same area in 2018 and 2019 of GF-6, respectively; (**c**,**d**) represent another set of tiles in 2018 and 2019 of GF-6, respectively.

Spectrum statistical characteristic values of each band for GF-6WFV are shown in Table 3. Among them, the near-infrared band (B4) has the largest standard deviation, followed by the red band (B3), the red-edge band 2 (B6), and the green band (B2), respectively. The blue band (B1) and the red-edge band 1 (B5) have a smaller standard deviation than the first three bands and the standard deviations of these two bands are similar, while the standard deviation of the purple band (B7) is the smallest. The standard deviation reflects the degree of discrete distribution of DN of each band. The higher the degree of dispersion, the greater the contrast of the image in this band, and the more abundant the information is. It is found that the near-infrared band contains the most abundant information, the spectral value distribution of pixels in this band is more dispersed, and the purple band has the smallest amount of information. The mean reflects the average size of the spectral value of each band. It shows that the near-infrared band (B4) is the highest, reaching 2778.32, and band 7 is the smallest.

**Table 3.** Statistical characteristic values of all bands of GF-6WFV.

|  | Min | Max | Mean | StdDev |
|---|---|---|---|---|
| Blue (1) | 729 | 3722 | 944.12 | 174.37 |
| Green (2) | 700 | 4094 | 1042.05 | 243.08 |
| Red (3) | 471 | 4094 | 820.12 | 303.88 |
| NIR (4) | 665 | 4095 | 2778.33 | 536.60 |
| RE1 (5) | 330 | 4088 | 701.96 | 180.05 |
| RE2 (6) | 381 | 4081 | 1448.52 | 264.46 |
| Purple (7) | 612 | 3064 | 713.01 | 77.99 |
| Yellow (8) | 437 | 4094 | 698.25 | 207.76 |

In summary, the new additional bands for GF-6WFV have the characteristics that the near-infrared band contains the largest amount of information, the red-edge band 1 contains less information than the red-edge band 2, and the purple band has the smallest amount of information.

After above analysis, we used GF-6WFV data to analyze the spectral values of different wavebands in each wave band and selected the features that can best express the change information. The features

selected in this study are: contrast, dissimilarity, correlation, entropy, and homogeneity. The extracted features of images are shown in Figure 6.

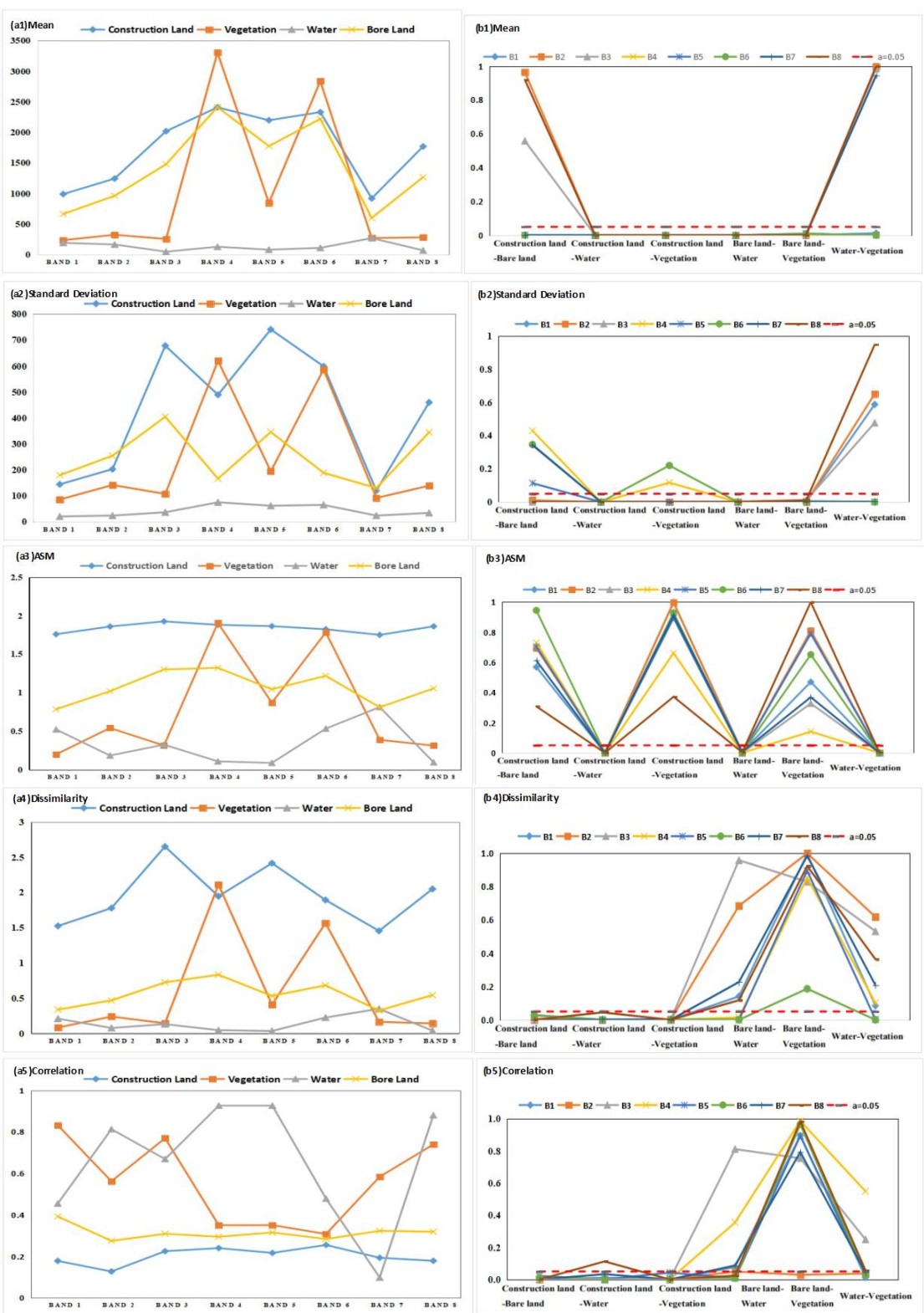

**Figure 6.** Comprehensive feature analysis for GF-6 WFV data ((**a1**~**a6**): spectral curves for each feature, (**b1**~**b6**): the ANOVA results for each feature, the *p*-value here is 0.05 (95% CI), which is represented as red dotted line in (**b1**~**b6**)).

The *F* value of each feature for the ANOVA (A 95% confidence interval, *F*0.05 = 5.117) is shown in Table 4.

**Table 4.** Statistical results of ANOVA.

|   | B1 | B2 | B3 | B4 | B5 | B6 | B7 | B8 |
|---|---|---|---|---|---|---|---|---|
| **Mean** | | | | | | | | |
| *F* | 83.7 | 65.57 | 92.28 | 174.34 | 67.43 | 150.93 | 53.98 | 83.26 |
| *p* | $6.64 \times 10^{-20}$ | $1.11 \times 10^{-17}$ | $8.04 \times 10^{-21}$ | $4.08 \times 10^{-27}$ | $3.18 \times 10^{-22}$ | $1.20 \times 10^{-25}$ | $6.22 \times 10^{-22}$ | $7.44 \times 10^{-20}$ |
| **Standard Deviation** | | | | | | | | |
| *F* | 46.95 | 47.45 | 31.25 | 27.21 | 37.15 | 27.12 | 20.81 | 25.44 |
| *p* | $1.48 \times 10^{-16}$ | $1.16 \times 10^{-16}$ | $8.20 \times 10^{-13}$ | $1.13 \times 10^{-11}$ | $3.14 \times 10^{-9}$ | $1.20 \times 10^{-11}$ | $5.99 \times 10^{-13}$ | $3.78 \times 10^{-11}$ |
| **Angular Second Moment (ASM)** | | | | | | | | |
| *F* | 10.25 | 17.58 | 15.87 | 17.69 | 15.33 | 16.62 | 11.64 | 10.17 |
| *p* | $1.76 \times 10^{-5}$ | $3.64 \times 10^{-8}$ | $1.38 \times 10^{-7}$ | $3.35 \times 10^{-8}$ | $5.83 \times 10^{-8}$ | $7.62 \times 10^{-8}$ | $2.19 \times 10^{-6}$ | $1.91 \times 10^{-5}$ |
| **Dissimilarity** | | | | | | | | |
| *F* | 16.81 | 14.83 | 16.38 | 15.19 | 21.59 | 24.74 | 9.76 | 9.81 |
| *p* | $6.58 \times 10^{-8}$ | $3.19 \times 10^{-7}$ | $9.24 \times 10^{-8}$ | $2.39 \times 10^{-7}$ | $2.11 \times 10^{-9}$ | $2.54 \times 10^{-10}$ | $4.29 \times 10^{-8}$ | $2.65 \times 10^{-5}$ |
| **Correlation** | | | | | | | | |
| *F* | 16.68 | 13.24 | 15.69 | 11.83 | 15.12 | 14.29 | 10.68 | 12.00 |
| *p* | $7.27 \times 10^{-8}$ | $1.21 \times 10^{-6}$ | $1.59 \times 10^{-7}$ | $4.15 \times 10^{-6}$ | $7.38 \times 10^{-8}$ | $5.00 \times 10^{-7}$ | $3.13 \times 10^{-8}$ | $3.56 \times 10^{-6}$ |

B1–B8 represent the eight bands of GF-6WFV, respectively.

As it can be seen from Table 4, the *F* value of each feature significance analysis is greater than *F*0.05, which shows that the difference between the five types of features selected in this paper between different types of features is significant. However, it is still uncertain whether the difference between any two feature types is significant. Therefore, the multiple comparisons were carried out for further analysis.

The results of spectral analysis and multiple comparisons for each band are summarized in Figure 6. In the spectral feature space, the difference in the Euclidean distance of the mean of various classes on B5 is the largest. In its corresponding significance analysis, the mean of B1, B4, B5, and B6 are significantly different between any two classes. The difference in standard deviations of Euclidean distance of all the classes on band B3, B4, and B5 is large, but in ANOVA, the difference between the construction land and bare land on these three bands is not significant, neither between the vegetation and water. The difference in the Euclidean distance of ASM of classes on B1, B2, and B8 is large, but any two classes in all bands have poor significance; The difference in the Euclidean distance of the dissimilarity of classes on B2 and B6 is significant. However, in the significance analysis, three pairs of classes are less significant on B2, while on B6, the significance between vegetation and bare land is poor. The difference in Euclidean distance of correlation of these classes on B2, B7, and B8 is large, but in the corresponding significance analysis, only B2 can be selected for the following change detection.

Among the four new bands in GF-6 WFV, B7 cannot be used as an effective feature in land change detection because of the little information contained. Although the amount of information in the B8 band is moderate, it means that it cannot provide sufficient effective information for change detection after significance analysis. The mean value of the two new red bands is significantly different between classes, but after the spectrum statistics, it can be found that the red-edge band 2 (B6) has a higher standard deviation than that of the red-edge band 1 (B5), indicates B6 has a larger image contrast.

*4.2. The Accuracy of LULC Change Detection Results*

After multi-scale segmentation, the feature vector of the object needs to be constructed through feature optimization. The analysis of variance in the previous section has selected the mean of bands 1, 4, 5, 6 of GF-6 WFV, respectively, the auto-correlation of band 2, and NDVI and NDWI to generate feature vectors. Then, through CVA and correlation analysis to obtain the difference intensity information between two images. Finally, determine the areas of change and non-change according to the threshold. The change detection results are shown in Figures 7 and 8.

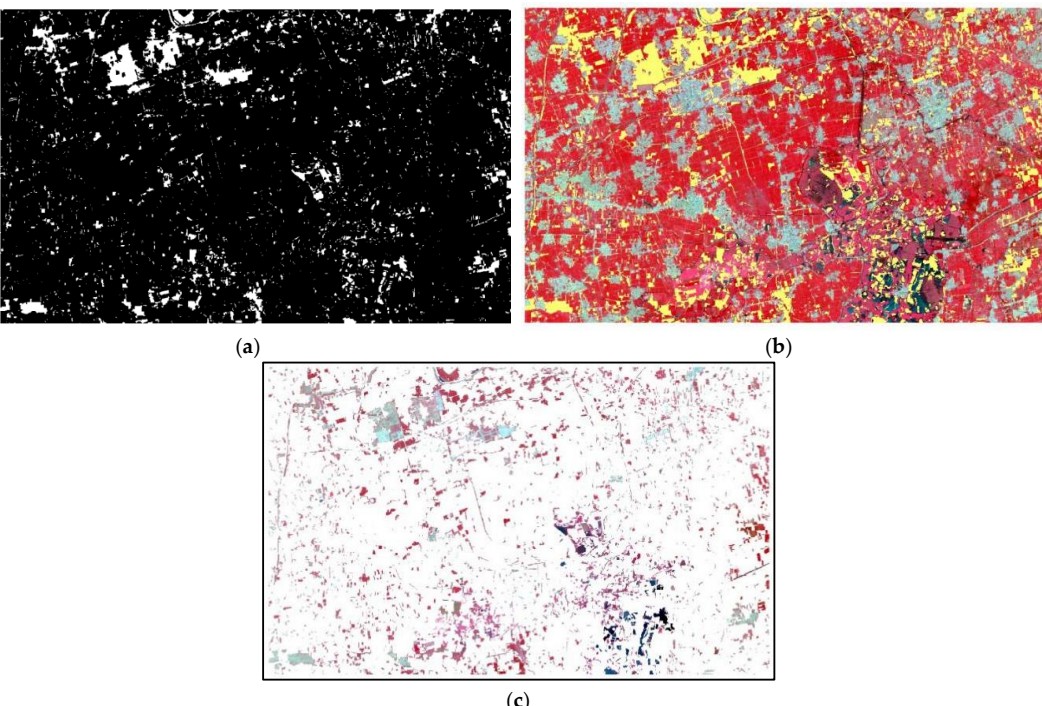

**Figure 7.** Change detection results by GF-6 WFV images of September of 2018 and 2019 (**a**) is binary (white represents changed areas); (**b**) is the superposition of the change detection results and the image (the yellow are the detected change areas; (**c**) is the base map of changed areas (white represents unchanged areas)).

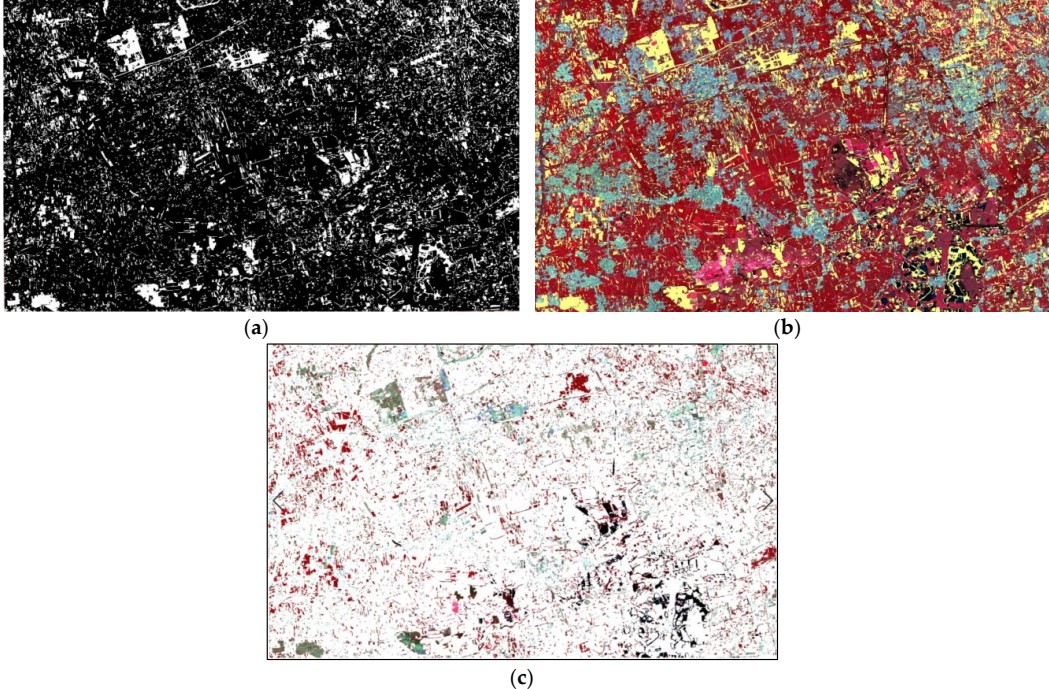

**Figure 8.** Change detection results by GF-1 WFV images of September of 2018 and 2019 (**a**) is binary (white represents changed areas); (**b**) is the superposition of the change detection results and the image (the yellow are the detected change areas; (**c**) is the base map of changed areas (white represents unchanged areas)).

In order to verify the accuracy of the model for GF-6WFV data, 300 samples were selected, including 133 samples without change and 167 samples with change. The selection of samples is carried out by the stratified sampling, randomly selecting the corresponding number of objects with change or without change individually, and then locate these objects through the generated ID numbers. The results are shown in Table 5.

**Table 5.** Confusion matrix of change detection using GF-6WFV images.

| Evaluation Index / Detection Results | Unchanged | Changed | Total |
|---|---|---|---|
| Unchanged | 119 | 19 | 138 |
| Changed | 14 | 148 | 162 |
| Total | 133 | 167 | 300 |

According to the calculations via the formulas, the commission error, omission error, and Kappa are 8.64%, 11.38%, and 0.7780, overall accuracy of 89%, respectively. For GF-6 WFV data with the resolution of 16 m, this accuracy can fully meet the application requirements of LULC change detection. Among the sample of the wrong and the missing, the proportion of each category is shown in Table 6. Among them, the change between bare land and vegetation has the highest probability of being incorrectly detected, reaching 42.4%, most of them are the farmland after harvest. In fact, such changes are often not regarded as real changes between classes. Due to the large proportion of farmland in the study area, the sample of farmland included in a random sampling sample will be higher than that of other land cover types, which leads to the largest proportion of farmland in the samples that were falsely detected and missed. Theoretically, the additional red-edge bands can provide more information for identifying changes in vegetation. For construction land, it has achieved good results for GF-6WFV data.

**Table 6.** Samples analysis for false/missing detection by GF-6WFV images.

| Class | Vegetation-Bare Land | Vegetation-Construction Land | Vegetation-Water | Bare Land-Water | Bare Land-Construction-on Land | Water-Construction Land |
|---|---|---|---|---|---|---|
| Number of samples | 14 | 4 | 2 | 1 | 11 | 1 |
| percentage | 42.4% | 12.1% | 6.1% | 3.0% | 33.3% | 3.0% |

It can be seen from Figure 7 that the Xiong'an New Area has undergone many changes in LULC from 2018 to 2019, and the types of changes mainly include construction land, vegetation, water bodies, and bare land. Moreover, the construction land changes account for a relatively large amount, including demolition and engineering facilities under construction. These changes are evenly distributed roughly throughout the whole study area.

*4.3. Comparison Results of GF-1WFV Images*

To verify whether the GF-6WFV data has an advantage in change detection, a comparison test was performed on GF-1WFV images in the same area and the same phase. The results are shown in Figure 8 and Table 7.

**Table 7.** The comparison of the change detection results for LULC by GF-6WFV and GF-1WFV data.

| Evaluation Index / Data Type | Overall Accuracy | Kappa | Commission Errors | Omission Errors |
|---|---|---|---|---|
| GF-1 WFV | 87% | 0.7351 | 13.21% | 9.58% |
| GF-6 WFV | 89% | 0.7760 | 8.64% | 11.38% |

The experiment found that the double-constrained change detection method can detect most of the change areas in the GF-1WFV images. Compared with the GF-6 data obtained on September 6, the GF-1 data obtained on September 24 obviously have many smaller blocky objects, most of which are harvested arable land, cause the late September coincides with the harvesting season of corn and sorghum. Due to the large amount of cloud interference in the data of GF-1 in the first ten days of September, we can only select the data in late September as a substitute in the experiment. As shown in Figure 8, there are many regular rectangular objects evenly distributed in the study area. Most of these are pseudo-changes caused by the harvest of cultivated land. In fact, the land cover type of the features themselves has not changed. To further verify the accuracy of the results, 300 samples were selected for comparative analysis, including 133 unchanged samples and 167 changed samples. Through comparison and calculation, the commission error is 13.21%, the omission error is 9.58%, Kappa is 0.7351, and the overall accuracy is 87%.

The result shows that the commission error of change detection under GF-1WFV data is significantly higher than that of GF-6WFV data using the same method, and the overall accuracy is reduced by 2%. Therefore, the accuracy of change detection that used the GF-6WFV data with the contained information of new additional bands is significantly improved. This indicates that these new red-edge bands contribute to the improvement of change detection accuracy and possess good potentials in the application of change detection.

## 5. Discussion

Through this research, it is necessary to discuss the following related issues.

In this paper, we only used the 16 m-resolution imageries of GF-6 WFV, not include other sensors' data of GF-6, such as panchromatic camera data and 8 m multi-spectral data for change detection of LULC. Although an accuracy of 89% is achieved, if the spatial resolution of the image can be improved, the accuracy of change detection should be higher. In order to fully test the performance of GF-6, the subsequent experiments can be considered, respectively. And the DCDM method used in the research is not an exclusive method proposed for GF series satellites, it is a general method that can be used for the detection of changes in various commonly used satellite data. In theory, if the appropriate features of objects are selected, the DCDM method can be suitable for any common used multi-spectral satellite imageries.

In terms of the change detection accuracy, the errors of registration preprocessing for different time-phase images, and vegetation phenology have some impacts on the final accuracy. The registration error is mainly caused by the change of the satellite parameters, the observation angle, topography and etc., which leads to the position of the same ground object in the images of different time phases to change. Although image registration is performed, this type of error cannot be completely eliminated, and we can try other methods to test whether such errors can be reduced. Moreover, taking the impact caused by phenology into consideration will help to detect the vegetation more accurately.

The experiments in this study are only for the change detection of bi-temporal images. For further validation, the more time phases for multi-temporal change detection should be tested and fully demonstrate the response of the new red-edge bands to change. This will be able to provide the advantages for resource monitoring or disaster assessment. And the automatic identification of the classes of LULC changes has not been done in this paper, it can be completed in the near future by learning the different types of features through machine learning methods.

## 6. Conclusions

The main conclusions of this study are as follows:

Firstly, through the spectral analysis and significance analysis of images of GF-6WFV, we found that its two new red-edge bands can provide effective information for change detection of LULC, while the purple and the yellow band cannot, and the addition of the new added two red-edge bands of

GF-6WFV increased the overall accuracy of 89% for change detection and improved by 2% compared with using the same temporal GF-1WFV data.

Secondly, the DCDM method based on change vector intensity and correlation coefficient can effectively detect the changes between different time-phase images for the same area. The object-oriented multi-scale segmentation method can more clearly describe the boundary of the changing area. It enables a more complete detection of the large change objects, as well as less omission of small change objects, making better use of the spectral and spatial information of images to obtain the changes. In addition, it can effectively reduce the generation of "salt and pepper" noise and increase the accuracy of change detection.

However, according to the validation, some narrow and long roads are incorrectly detected due to the errors caused by the registration, which affects the overall accuracy.

In summary, it can be concluded from the study that GF-6, as the first high-resolution satellite for precision agricultural observation in China, can meet the accuracy requirements of change detection of LULC with the combination of the new red-edge bands, and there is a good potential in monitoring resource and environment, for example, agriculture and forestry, and disaster mitigation, etc. This study also provides a theoretical support for the in-depth applications in related fields.

**Author Contributions:** All authors contributed in a substantial manner to the manuscript. Conceived, designed, and performed the research and wrote the manuscript, J.Y.; made contributions to conceive, and design the research, data analysis, and wrote some parts of the manuscript, Y.L.; contributed the model and results analysis and the conclusion, Y.R.; contributed to the programming, H.M.; contributed to the data preprocessing, Y.J., D.W. and L.Y. All authors discussed the basic structure of the manuscript. All authors have read and agreed to the published version of the manuscript.

**Funding:** This study was funded by the major projects of High resolution Earth Observation System of China (30-Y20A07-9003-17/18).

**Acknowledgments:** We would like to thank the staffs who provided the data and thank the reviewers for their comments and suggestions.

**Conflicts of Interest:** The authors declare no conflict of interest.

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
