# Peer review of "Application Study on Double-Constrained Change Detection for Land Use/Land Cover Based on GF-6 WFV Imageries"

_remotesensing, doi:10.3390/rs12182943_

Round 1

Reviewer 1 Report

The manuscript, entitled "Application Study on Double-constrained Change Detection for Land Use/Land Cover Based on GF-6 WFV Imageries" (by Yu J. et al.) focuses on the combined use of  GF-6 WFV imagery data and the Double-constrained Change Detection Method (DCDM) for land use/cover (LULC) detection in a region of China. One of the main objectives of the study was to examine the possibility of production of better change results due to the better technical characteristics of GF-6 WFV than those of GF-1 WFV. For this reason, a comparison of the results derived from the imagery data of these different satellite sensors was applied in terms of accuracy metrics.

The manuscript deals with a topic of international interest which makes it suitable for publication. The proposed methodology and its output results are certainly valuable. Nonetheless, it has several issues that must be solved in order to be acceptable for publication in the "Remote Sensing" journal. Below I report some general remarks, as well as some minor comments and suggestions so that the authors take them under consideration.

General remarks

- Regarding the structure of the manuscript, the "Data" section includes a brief description of the study area. More information about the study area should be provided, focusing more on its predominant land use/cover types. Hence, either create a separate "Study area" section or at least rename the existing section as "Study area and Data". Moreover, the "Discussion and Conclusions" section is too short. Its enrichment is necessary; among others, there is none mention about the usefulness of the produced results.

- Regarding the applied methodology, it is not very clear to the readers the empirical correlation between the implementation of CVA and correlation coefficient methodological components, defined together as DCDM method, and the used imagery data. I think that this is very important since the main objective of the study is the change detection. In addition, the generation of feature vectors based on, among other, the spectral indices of NDVI and NDWI is mentioned without providing none information about their estimation (e.g. the relative Equations). The confidence level based on which the ANOVA-based statistical differences are recognized as significant is totally missing. For instance, is this 95% due to the value of 0.05?

- Regarding the produced results, they mainly indicate the changed/unchanged parts of the study area, but not at all the types of changes (from one land use/cover type to another); which constitutes very important information in studies related to LULC change detection. Since the authors mention that in CVA "the angle of the change vector represents the type of change", why not they dealt with this in order to also provide the relevant information. Instead of that, they are limited just to a mention like "From Figures 7, it can be seen Xiong'an New area has undergone many changes in LULC from 2018 to 2019, and the types of changes mainly include construction land, vegetation, water bodies, and bare land". However, this can be considered as an unconfirmed finding, particularly since none "picture" is provided to the readers presenting the LULC conditions (i.e. types) in the study area for the two different examined temporal phases.

- Regarding the accuracy of the results, it is not clear how the used 300 samples (as reference data?) were selected, and mainly derived (e.g. by the interpretation of the imagery data or by another procedure?). In which format they also were?

Minor comments

No line numbers in the manuscript making the review work difficult to the Reviewers.

The citation of Figures in the main text either is missing or causes confusion to the readers due to incorrect correlation between them.

Some more specific comments are provided in the attached PDF file.

Reviewer 2 Report

Overall, the paper is well presented. It provides an interesting account of the methods used. However, more could be done.

1) The introduction does not provide enough theoretical support for the subject. It is a very general review and it is taking the topic from a too-long distance;

2) Please, clarify and what are the innovative contributions to science. Highlight the aspects in which your research departs from the existing literature;

3) The objectives should be clearly stated in the introduction;

4) I would encourage the authors to directly compare the GF-6 WFV Imageries with other common satellite images as this would be very useful to the broad community. What are its advantages?

5) Methodology needs more explanation of issues and alternative approaches to overcome.

6) Study area: Figure 1 - I strongly recommend reworking the cartographic appearance.

7) The authors should be clearer about why choosing this particular study area. The contextualization is not enough. All elements that could help the reader better picture what looks like the area we are talking about.

8) The discussion should be separated from the conclusion. And in fact, there is no real discussion in the section ‘discussion and conclusion’.

9) You should recognize 'how the methods and the outcomes could be applicable elsewhere'. This will help boost the potential international readership of the manuscript. 

10) The conclusion ends too fast. Please, highlight the main achievements of the research.

Round 2

Reviewer 1 Report

Dear Authors,

Thank you for taking into your account all my comments/suggestions.

For this reason, I propose the acceptance of your manuscript in its present form.

Congratulations for your good work!

Reviewer 2 Report

Thank you for the changes that you have made. However, the discussion section is still far from being a real discussion. You should reformulate it.